# Navigating the Immune Maze: Pioneering Strategies for Unshackling Cancer Immunotherapy Resistance

**DOI:** 10.3390/cancers15245857

**Published:** 2023-12-15

**Authors:** Liqin Yao, Qingqing Wang, Wenxue Ma

**Affiliations:** 1Key Laboratory for Translational Medicine, The First Affiliated Hospital, Huzhou University, Huzhou 313000, China; 2Institute of Immunology, Zhejiang University School of Medicine, Hangzhou 310058, China; wqq@zju.edu.cn; 3Department of Medicine, Moores Cancer Center, Sanford Stem Cell Institute, University of California San Diego, La Jolla, CA 92093, USA

**Keywords:** cancer immunotherapy, resistance, tumor microenvironment, combination therapies, immune checkpoint targets, adoptive cell therapies, cancer vaccines, personalized medicine

## Abstract

**Simple Summary:**

Cancer treatments have made remarkable advances with the introduction of immunotherapy, which recruits the body’s immune system to fight cancer. Despite these advancements, cancer can sometimes develop resistance to such treatments, diminishing their effectiveness. Our research is focused on the early detection of signs that indicate a cancer’s resistance to immunotherapy, enabling physicians to swiftly alter treatment approaches and improve the chances of patient recovery. We are particularly keen on identifying distinct markers in tumors that indicate this resistance. To achieve a deeper understanding, we utilized scaled-down models of patient tumors, including organoids and xenografts, in laboratory studies. Our goal was to discover innovative methods to combat treatment resistance, potentially enhancing patient care and providing valuable insights for ongoing cancer research.

**Abstract:**

Cancer immunotherapy has ushered in a transformative era in oncology, offering unprecedented promise and opportunities. Despite its remarkable breakthroughs, the field continues to grapple with the persistent challenge of treatment resistance. This resistance not only undermines the widespread efficacy of these pioneering treatments, but also underscores the pressing need for further research. Our exploration into the intricate realm of cancer immunotherapy resistance reveals various mechanisms at play, from primary and secondary resistance to the significant impact of genetic and epigenetic factors, as well as the crucial role of the tumor microenvironment (TME). Furthermore, we stress the importance of devising innovative strategies to counteract this resistance, such as employing combination therapies, tailoring immune checkpoints, and implementing real-time monitoring. By championing these state-of-the-art methods, we anticipate a paradigm that blends personalized healthcare with improved treatment options and is firmly committed to patient welfare. Through a comprehensive and multifaceted approach, we strive to tackle the challenges of resistance, aspiring to elevate cancer immunotherapy as a beacon of hope for patients around the world.

## 1. Introduction

Cancer immunotherapy heralds a promising revolution in the realm of oncological treatments. This groundbreaking approach, rooted in historical milestones like “Coley’s toxins” [1] and, later, the identification of cytotoxic T-lymphocyte-associated antigen 4 (CTLA-4), has consistently showcased the potential to redefine cancer treatment paradigms [2,3,4]. As we deepened our understanding of tumor antigens and immune–tumor interactions in the latter half of the 20th century, the emergence of agents targeting CTLA-4, programmed cell death protein 1 (PD-1), and programmed death-ligand 1 (PD-L1) pathways marked significant successes in treating a range of malignancies [5,6,7]. Additionally, personalized strategies, such as chimeric antigen receptor (CAR) T-cell therapies, offer compelling efficacy, particularly in hematological malignancies [8,9,10]. The scope of cancer immunotherapy has since broadened, delving into influencing factors like the tumor microenvironment (TME) and even the gut microbiome to amplify therapeutic impact [11,12].

Despite these advances, resistance to immunotherapy presents a formidable barrier, emerging from innate tumor characteristics and adaptive changes in the genetic and proteomic landscape [13]. At the heart of this challenge lies the TME, which harbors elements like regulatory T cells (Tregs) and certain cytokines that shield tumor cells, allowing them to cleverly sidestep immune detection [14,15,16].

Our objectives are to dissect the complexity of immunotherapy resistance, evaluate both primary and secondary mechanisms, and consider the profound influence of genetic, epigenetic, and environmental factors [17]. We spotlight emerging strategies to overcome resistance and highlight the necessity of an integrated approach involving real-time monitoring, precision analytics, and patient-centered care [18]. By addressing these challenges head-on, we aim to advance the efficacy of cancer immunotherapy, reinforcing its position as a cornerstone of modern cancer care.

Through navigating the intricate landscape of resistance, we present insights into both established and novel strategies to outmaneuver the adaptive nature of tumors [19]. This review encapsulates the critical need for adaptability in treatment approaches, the ongoing quest for data-driven precision in patient-focused care, and the overarching potential of immunotherapy to redefine the future of cancer treatment [20,21,22].

## 2. The Immune Maze: Understanding the Complex Landscape

At the heart of the challenges presented by immunotherapy lies a deep-rooted, intricate interplay between the immune system and cancerous tumors. Grasping this landscape is pivotal to addressing the ever-evolving complexities of immunotherapy resistance [23,24]. To embark on this journey, it is crucial to recognize the distinctions between primary and secondary resistance and the multifarious mechanisms that underlie them [25].

Primary resistance: Innate to certain tumors, primary resistance emerges due to various factors that hinder the immune system’s capability to detect and counteract tumor cells. Some tumors are devoid of the critical antigens essential for immune recognition, rendering them less amenable to immunotherapeutic strategies [26,27]. Another dominant culprit is the immunosuppressive TME, characterized by a plethora of inhibitory factors and cells that dampen immune responses [28,29].

Consequently, secondary resistance develops as a backlash to therapeutic interventions. This form of resistance revitalizes tumor growth even after an initial successful response to immunotherapy including nivolumab (a PD-1 inhibitor) and ipilimumab (a CTLA-4 inhibitor) [30]. The driving forces behind this resistance span a spectrum from the genetic evolution of the tumor, which can lead to the modification or loss of previously identifiable antigens, to dynamic modifications to the TME, such as the amplification of immunosuppressive molecules or the influx of inhibitory cells [26,31,32].

Building on this, recent discoveries in the field have shed light on crucial aspects of immunotherapy resistance. Cutting-edge research has delved into the genetic and epigenetic blueprints of tumors. It has been shown that genetic modifications can recalibrate a tumor’s antigenic composition, impeding its visibility to immune cells [27,33,34,35]. Moreover, epigenetic shifts can mute genes vital for immune detection without altering the DNA structure or can modify how the tumor communicates with the surrounding immune framework [36,37,38].

Simultaneously, within the TME are distinct cellular entities that have gained prominence. These include Tregs, myeloid-derived suppressor cells (MDSCs), and tumor-associated macrophages (TAMs), which play cardinal roles in dampening immune activity and forming a protective bulwark around tumors [39,40,41]. Current research endeavors are evaluating their potential as resistance biomarkers, offering a glimpse into therapeutic trajectories [42,43].

Another pivotal aspect is the TME hypoxia [44,45]. Rapid tumor growth often surpasses its vascular supply, instigating hypoxia, which in turn sparks resistance pathways [44,46,47]. This oxygen deficiency is correlated with elevated PD-L1 expression, which mutes T-cell responses, facilitating tumor evasion [48,49].

Furthermore, the interplay between tumors and major histocompatibility complex (MHC) molecules is gaining traction [9,50]. MHCs are paramount in displaying tumor-specific peptides on the tumor surface for the T-cell detection [27,51,52]. Tumors have been found to employ evasion techniques, such as downregulating MHC expression or tweaking antigen-processing systems [27,53].

On a related note, immune checkpoints continue to be a focal point in the resistance discourse [54,55]. Often regulators in the immune system, these checkpoints are manipulated by tumors to serve as barriers against immune onslaughts [56,57]. Contemporary treatments, especially checkpoint disruptors, aspire to dismantle these barriers, amplifying immune responses against malignancies [7,58,59]. The latest clinical trials are unraveling the effectiveness of and obstacles to bypassing checkpoint-triggered resistance [60,61,62,63].

In summary, a profound understanding of the intricacies of immunotherapy resistance, its genesis, current revelations, and the TME’s role is fundamental in forging ahead with innovative strategies to subvert these hurdles. Subsequent sections provide a deeper exploration of these tactics.

## 3. Frontline Foes: Decoding the Architects of Immunotherapy Resistance

The TME serves as a dynamic milieu, evolving continuously and influencing the efficacy of cancer immunotherapies [64]. Key cytokines, notably transforming growth factor beta (TGF-β) and IL-10, are pivotal in modulating the TME, orchestrating immunosuppressive signals that underpin tumor resilience against therapeutic strategies.

Tregs are essential players within the TME, possessing the capability to subdue robust immune responses, particularly from formidable cells like cytotoxic T cells (CTLs) [65,66,67]. This suppression presents formidable challenges for immunotherapies, with Tregs secreting TGF-β and IL-10 to augment their inhibitory functions [68,69].

MDSCs further complicate the TME dynamics. These immune cells exacerbate the suppressive atmosphere, inhibiting CTLs and natural killer (NK) cells, thus limiting their tumor-fighting abilities [43,70]. They excel in restraining CTLs and NK cells, thus curtailing the NK cells’ tumor-eradicating capabilities [43,71,72]. Additionally, the MDSCs foster Treg proliferation, intensifying the suppressive milieu [73,74].

TAMs, with their versatile roles, are noteworthy contributors to the TME. Their ability to transition between M1-like (TAM1) and M2-like (TAM2) states plays a significant role in the balance between tumor defense and progression [75,76]. While TAM1 cells act aggressively against cancer cells, TAM2 cells encourage a suppressive environment, promoting tissue repair and angiogenesis, as well as safeguarding tumors from immune attacks [77,78,79].

Tumor-associated neutrophils (TAN) also differentiate into two major phenotypes within the TME. While TAN1 cells inhibit cancer progression, TAN2 cells support tumor growth, underscoring the multifaceted interactions within the TME [80,81].

Other factors, like rapid tumor growth leading to hypoxic conditions, activate various resistance mechanisms [82,83]. This includes the upregulation of immune checkpoint molecules such as PD-L1 on tumor surfaces, hindering T-cell functionality [84,85]. Hypoxia-triggered signaling pathways further deepen the TME’s suppressive nature [45,86].

Cancer cells also deploy evasion strategies, manipulating MHC molecules to reduce their visibility to the immune system [87,88]. Despite the promise of immune checkpoint inhibitors (ICIs), challenges remain in terms of assuring sustained outcomes and managing emergent resistance [7,89,90].

In closing, a profound grasp of these pivotal agents within the TME is paramount for charting successful strategies against the immunotherapy resistance [91]. As the research community continues its quest, the hope is to modulate these elements, enhancing the potency of the cancer immunotherapy [91,92,93]. By appreciating the TME’s intricacies, we inch closer to reshaping therapeutic outcomes and offering renewed hope to countless patients.

Figure 1 below provides a schematic representation of the intricate cellular interactions within the hypoxic TME, highlighting the key players involved in immunotherapy resistance.

## 4. Pioneering Strategies to Overcome Resistance

Cancer immunotherapy, while promising, is often hindered by the development of resistance. Several innovative strategies have been developed to address this, each designed to improve patient outcomes and enhance treatment efficacy.

### 4.1. Combination Therapies

Combination therapies represent a multi-pronged attack against cancer, targeting different aspects of tumor biology. These therapies may combine agents that halt tumor growth with those that boost the immune response. Despite the potential for increased toxicity, the benefits often outweigh the risks, necessitating careful patient management [94,95,96].

### 4.2. Tumor Microenvironment (TME)

Strategies that modify the TME aim to disrupt the supportive network of the tumor, including alterations in blood flow and stromal cell inhibition. Such interventions highlight the TME’s critical role in cancer therapy [97,98,99,100,101,102].

### 4.3. Emerging Immune Checkpoints

New research is focused on uncovering and targeting novel immune checkpoints that tumors exploit to evade immune detection. Agents targeting the ITIM domain (TIGIT), T cell immunoglobulin and mucin-domain-containing-3 (TIM-3), and lymphocyte activation gene-3 (LAG-3) are under investigation for their therapeutic potential [103,104].

### 4.4. Enhancing Immunotherapy with Oncolytic Viruses

Oncolytic viruses are emerging as a novel countermeasure to immunotherapy resistance. These viruses are engineered to selectively infect and destroy cancer cells while also modulating the immune environment to reverse resistance mechanisms. For example, the oncolytic virus VSV-GP, when combined with PD-1 inhibitors, has been found to effectively kill tumor cells. It also encourages the maturation of DCs and the influx of T-cells into the tumor milieu, which are crucial steps in reigniting the immune system’s attack on the cancer [105].

Furthermore, clinical trials, such as one led by Chesney et al., have revealed that T-VEC, an oncolytic virus derived from the herpes simplex virus, can significantly enhance treatment outcomes for melanoma patients, especially when administered in conjunction with ICIs [106]. This dual approach not only targets the tumor directly, but also reactivates the patient’s immune response against the tumor, providing a two-pronged attack against cancer resistance.

These developments signify a stride forward in integrating oncolytic virotherapy into the arsenal of immunotherapeutic strategies. By continuing to leverage these biological agents, researchers aim to unlock new pathways to overcome resistance and maximize the therapeutic potential of cancer immunotherapy.

### 4.5. Cell Therapy (ACT)

ACT personalizes treatment by using the patient’s immune cells, like TILs or chimeric antigen receptor (CAR)-T cells, to combat cancer. While effective in blood cancers, its application in solid tumors is an active area of research [107,108,109,110].

### 4.6. Cancer Vaccines

Cancer vaccines aim to prime the immune system to recognize and attack tumors, with DC and viral vector vaccines leading the way. This strategy is part of a broader effort to induce durable immune responses against cancer [111,112,113,114].

### 4.7. Navigating Medication-Induced Resistance in Immunotherapy

The interplay between certain medications and cancer immunotherapy is complex and can inadvertently contribute to treatment resistance. Corticosteroids, which are commonly prescribed to alleviate the side effects of immunotherapy, may inadvertently suppress the immune response, reducing the efficacy of treatments like ICIs [115,116]. Additionally, chemotherapeutic agents, while targeting cancer cells, may also inadvertently modify the immune environment in a way that fosters resistance [117,118]. This alteration in the immune landscape can hinder the immune system’s ability to effectively recognize and attack tumor cells.

Moreover, the use of antibiotics has been linked to disruptions in the gut microbiome, an emerging factor in the modulation of immunotherapy responses [119]. The gut microbiome plays a crucial role in maintaining a balanced immune system, and its disturbance may impact the success of immunotherapeutic strategies.

Furthermore, kinase inhibitors, used in targeted therapies, might alter critical signaling pathways that are essential for the activation and function of immune cells, contributing to a resistance scenario [120,121]. Such unintended effects underscore the necessity for clinicians to carefully consider the full spectrum of a patient’s medication regimen when administering immunotherapy.

By comprehensively understanding these drug interactions and their implications, medical professionals can devise strategies to avoid or counteract the resistance-inducing effects of these drugs. This may involve adjusting dosages, sequencing treatments, or selecting alternative therapeutic agents to maintain the robustness of the immune response [122].

Integrating advanced strategies that account for drug-induced resistance with conventional cancer therapies represents a significant step toward a new era in cancer treatment. This multifaceted approach emphasizes the need for continuous research and adaptation to refine immunotherapy regimens, ensuring they remain potent against cancer while respecting the patient’s overall well-being and minimizing unintended resistance [17,123].

Figure 2 below provides a visual representation of the different immunotherapeutic agents and their specific targets within the tumor microenvironment, illustrating the mechanisms by which they exert their effects.

### 4.8. Integrated Strategies for Overcoming Resistance

To surmount the challenges presented by resistance to immunotherapy, an integrated approach is necessary. This involves not only the combination of therapeutic modalities but also the development of new agents that can tackle the evolved defense mechanisms of tumors. Precision medicine plays a crucial role in this, with targeted therapies designed to counteract specific pathways of resistance identified in a patient’s tumor profile [17]. Adopting personalized treatment regimens based on molecular diagnostics and patient-derived models, such as organoids and xenografts, is showing promise in enhancing treatment efficacy and reducing toxicity [123]. Furthermore, the implementation of real-time monitoring systems and predictive biomarkers facilitates a more responsive approach to immunotherapy adjustments [124,125]. The future of overcoming immunotherapy resistance lies in the synergy of these innovative strategies, each contributing a piece to the complex puzzle of cancer treatment [126].

In the following section, we provide an overview of pioneering strategies in cancer immunotherapy. Table 1 summarizes these strategies, including their approaches, key components, benefits, drug examples, and supporting references.

To wrap up this exploration, the integration of these advanced strategies with traditional therapies offers a multifaceted approach to overcoming immunotherapy resistance, signaling a new era of hope for cancer treatment [129,130].

## 5. Recent Insights and Developments in Overcoming Immunotherapy Resistance

The endeavor to unravel and overcome resistance in cancer immunotherapy has uncovered significant genetic and epigenetic influences that affect patient outcomes [91,131,132,133].

### 5.1. Genetic Alterations and Immunotherapy Resistance

The emergence of resistance to immunotherapy due to genetic alterations within cancer cells is a major concern that complicates treatment outcomes. These mutations can significantly alter the immune system’s ability to recognize and destroy cancer cells. One of the key genetic changes involves mutations in the beta-2-microglobulin (B2M) gene, a critical component of the major histocompatibility complex (MHC) class I molecules. The MHC class I molecule presents tumor antigens to T cells, and any disruption in this pathway, as caused by B2M mutations, can lead to ineffective T cell-mediated tumor cell lysis [134,135].

Moreover, the Janus kinase (JAK) pathway, which includes the genes JAK1 and JAK2, plays a pivotal role in immune response signaling [136]. Mutations in these genes can have profound effects on the efficacy of immunotherapies. Shen et al.’s investigation into JAK1/JAK2 alterations revealed that such mutations can result in resistance to PD-1 blockade therapies by impairing the interferon signaling pathway, which is vital for the activation of the immune response against tumor cells [137].

Additionally, research indicates that alterations in the neoantigen landscape of cancer cells, due to genetic mutations, can influence the responsiveness to immunotherapy. The mutational burden and the quality of the neoantigens presented can either enhance or diminish the therapeutic efficacy, as the immune system may or may not recognize these neoantigens as targets [138,139].

These genetic alterations underscore the need for comprehensive genomic profiling of tumors to anticipate and overcome resistance mechanisms. By understanding and mapping these genetic changes, clinicians can personalize immunotherapy approaches, potentially restoring the sensitivity of cancer cells to treatment and improving patient prognosis.

### 5.2. Epigenetic Dynamics and Their Role in Resistance

The regulatory landscape of epigenetic modifications is significant in immunotherapy resistance, profoundly affecting gene expression and the immune detection of tumors. DNA methylation, which adds a methyl group to DNA and often leads to gene silencing, has been implicated in immune evasion. Mehdi et al. [140] have identified that hypermethylation of the promoter regions of Th1-type cytokine genes can result in the suppression of crucial immune signaling pathways. This hypermethylation effectively reduces the expression of cytokines necessary for a robust anti-tumor immune response, thus facilitating tumor cells’ escape from immune surveillance [141].

Histone modifications, another crucial aspect of epigenetics, involve changes to the proteins around which DNA is wound. Histone acetylation and deacetylation, controlled by histone acetyltransferases (HATs) and histone deacetylases (HDACs), can alter the accessibility of DNA to transcription machinery. Aberrations in HDAC activity have been linked to the repression of tumor suppressor genes. For example, overactivity of HDACs can lead to the tight winding of DNA around histones, effectively “hiding” tumor antigens from immune cells and contributing to resistance to immunotherapies such as checkpoint inhibitors [141,142].

Specific treatments, like the DNA methyltransferase inhibitors azacitidine and decitabine, have been shown to induce these epigenetic changes. They can enhance the effectiveness of immunotherapy by altering the expression of cancer/testis antigens and MHC molecules, heightening tumor immunogenicity [34,143]. However, they can also trigger immune evasion, necessitating a nuanced approach to their use in conjunction with immunotherapies [144].

Histone deacetylase inhibitors, such as vorinostat and romidepsin, have dual roles. While they can increase antigen presentation, they have also been implicated in promoting regulatory T-cell functions, which could dampen the immune response [145,146]. This highlights the delicate balance required when integrating epigenetic therapies with immunotherapy and underscores the need for further research to optimize these combinations.

### 5.3. The Microbiome’s Influence on Immunotherapy Efficacy

The interplay between the gut microbiome and the efficacy of cancer immunotherapy is a an intensively researched topic. The diverse community of microbes residing in the gastrointestinal tract exerts a substantial influence on the body’s immune responses, with significant implications for the effectiveness of immunotherapeutic agents.

In a landmark study by Derosa et al., researchers identified that the presence of specific gut bacteria, such as Akkermansia muciniphila, significantly improved the efficacy of PD-1 inhibitors. This microbe appeared to bolster the host immune system’s capacity for tumor surveillance, potentially by maintaining mucosal integrity or enhancing immune cell activation, thus increasing the effectiveness of immunotherapies [147]. Such findings have led to the proposal that the gut microbiome could serve as a predictive biomarker for immunotherapy responses, and through interventions such as diet or probiotics, could be adjusted to improve clinical outcomes.

Conversely, antibiotic use can disrupt the delicate balance of the gut microbiome, with studies like those conducted by Patel et al. demonstrating negative impacts on the efficacy of immunotherapies. Antibiotics may diminish beneficial bacteria, impair immune function, and lessen the host’s response to PD-1 inhibitors, highlighting the need for careful consideration of antibiotic use during immunotherapy [148].

This emerging research area has spurred interest in probiotics and fecal microbiota transplantation (FMT) as methods to modulate the gut microbiome favorably. Ongoing clinical trials are exploring the potential of these interventions to modulate the gut microbiome in order to improve the patient response rate to cancer immunotherapy [149,150].

Overall, a growing body of evidence supports the notion that therapeutic modulation of the microbiome could serve as an adjunct to enhance the efficacy of immunotherapy and reduce resistance. Ongoing research into microbiome-based adjuvants holds promise for refining the management of cancer through these novel interventions.

## 6. Clinical Implications and Translational Approaches

The recognition and early identification of biomarkers indicative of resistance is pivotal in optimizing cancer treatment protocols. Biomarkers, such as high PD-L1 expression or a significant tumor mutational burden (TMB), as well as genetic alterations like JAK1/2 mutations, are at the forefront of predicting and countering immunotherapy resistance [151]. These biomarkers not only facilitate diagnosis, but are also vital for the creation of targeted strategies that preemptively confront specific resistance pathways [152].

Translational research tools like patient-derived organoids (PDOs) and xenograft models (PDX) are instrumental in applying preclinical findings to clinical treatment design. For instance, PDOs derived from colorectal cancer patients have been utilized to evaluate the efficacy of novel drugs, replicating the complex cellular environment of the originating tumor [153,154]. These studies have led directly to clinical trials and adjustments to treatment regimens, exemplifying how PDOs can significantly influence therapeutic planning and patient management.

In the vanguard of translational research, PDX models stand out for their direct impact on clinical decision-making. By engrafting human tumor tissues into immunodeficient mice, PDX models maintain the tumor’s intrinsic heterogeneity, providing insights into the tumor’s response to new treatments. These models have significantly advanced our understanding of resistance mechanisms, guiding the design of clinical trials aimed at targeted resistance pathways.

For instance, PDX research has led to the discovery of alternative immune checkpoints and changes in antigen presentation, shaping the development of combination therapies and influencing clinical treatment modifications. Such studies have also identified biomarkers predictive of treatment response, allowing for the adaptation of clinical protocols [155]. 

A key example of the impact of PDX models is their use in pinpointing specific genetic mutations that confer resistance to standard therapies. Insights gained from PDX studies have informed the enrollment of patients in trials for new targeted agents, leading to improved outcomes. These translational models are thus integral to the evolution of personalized medicine, enhancing the specificity and adaptability of cancer therapies [155].

PDX models, together with PDOs, enhance therapeutic planning by replicating the complex tumor environment, thereby offering a dynamic platform for drug evaluation and the development of personalized treatment regimens [153,154].

The synergy between clinical acumen and advanced translational models is reshaping cancer therapy, increasing the precision of the current treatments, and paving the way for innovative strategies to navigate the complexities of immunotherapy resistance. This integrated approach is set to refine patient care, promising a future where cancer treatment is as personalized as it is effective.

## 7. Future Perspectives in Immunotherapy

The future of immunotherapy is illuminated by advancements across varied disciplines, seamlessly integrating cutting-edge technologies poised to redefine oncological breakthroughs.

At the vanguard of these advancements, the integration of artificial intelligence (AI) and machine learning offers the capability to decipher vast genetic and proteomic datasets [156,157,158]. While this technological leap revolutionizes personalized immunotherapy by predicting tumor behavior and resistance mechanisms, as well as enabling real-time patient monitoring, it also brings forth challenges. For instance, ensuring the privacy and security of patient data processed by AI becomes paramount. Moreover, the algorithms’ decision-making processes require transparency, especially when used to make clinical recommendations. Ethical considerations arise, questioning the extent of reliance on AI for treatment decisions and potential biases embedded within the algorithms.

Nanotechnology, emphasizing nanoparticles, holds significant potential to enhance the immunotherapy [8,50,52,159,160,161]. Its ability to deliver drugs precisely to tumor sites and fine-tune immune responses charts the path for groundbreaking strategies. These include modifying the TME to impede tumor growth, optimizing nutrient dynamics within the TME, and propelling the development of neoantigen vaccines. However, the use of nanoparticles raises concerns regarding long-term safety, potential off-target effects, and their interactions with the body’s natural systems. Ethical discussions also surround the equitable distribution of such advanced treatments and the potential high costs associated with them.

Tumor epigenetics is a rising domain, with research directed toward harnessing epigenetic modulators to manipulate gene expression patterns. This tactic could potentially combat immunotherapeutic resistance, thus diversifying treatment avenues.

Simultaneously, telemedicine platforms are bridging geographical chasms, ensuring that specialized care becomes universally accessible [162]. Such platforms empower individuals in regions with constrained specialty resources to receive optimal treatment recommendations. The prevailing transformative phase in immunotherapy flourishes with interdisciplinary collaboration. Disciplines like genetics, immunology, bioengineering, and sociology coalesce, exemplified by the amalgamation of genomic sequencing, microfluidic technologies, and 3D tumor modeling to sharpen therapeutic strategies.

In summation, the dynamic realm of immunotherapy intertwines an array of disciplines, pioneering technologies, and global partnerships. The forthcoming epoch promises unmatched precision and flexibility, as well as a rejuvenated wave of oncological innovations, albeit not without its challenges and ethical dilemmas.

## 8. Conclusions

Throughout our journey into the complex landscape of immunotherapy, we confronted a myriad of challenges and opportunities. The foremost among these was the issue of immunotherapy resistance. While such challenges might seem daunting, they also serve as gateways to novel innovations. Our increasingly profound comprehension, bolstered by advancements in AI, nanotechnology, and epigenetics, is propelling us toward solutions that were once considered beyond reach.

Immunotherapy heralds a paradigm shift in oncological treatments, emphasizing the body’s intrinsic defenses against malignancies. Yet, the ever-present shadow of resistance reminds us of the continuous need for exploration, adaptation, and innovation. It is the collective endeavors of researchers, clinicians, and pioneers across disciplines that underpin the remarkable breakthroughs we witness today. These efforts inch us closer to the overarching goal: to overcome cancer resistance and elevate patient outcomes.

However, like all scientific pursuits, our research has its confines. Future studies might focus on deeper dives into molecular mechanisms, patient-specific factors, or even socio-economic considerations that could influence resistance. Expanding on these areas would undeniably enrich our understanding.

In summary, our journey through the complexities of immunotherapy resistance is continuous, but the advancements made signal a hopeful future. Here, cancer treatments are envisioned to be not only more personalized and powerful, but also characterized by fewer adverse effects. The crux of this progress lies in persistent research, international cooperation, and a steadfast commitment to revolutionizing the story of cancer treatment.

## Figures and Tables

**Figure 1 cancers-15-05857-f001:**
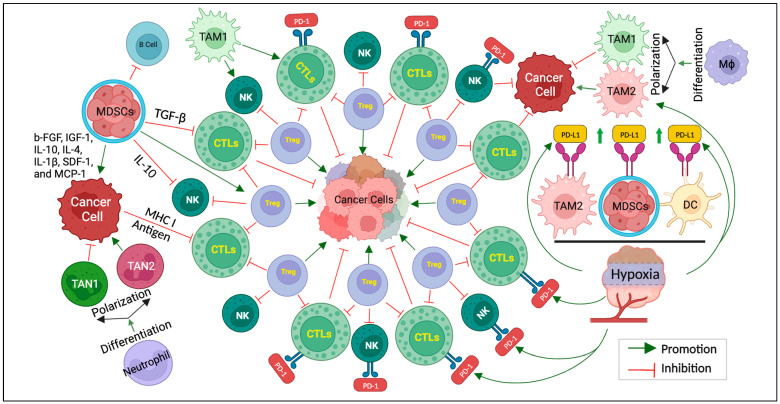
The keys to overcoming immunotherapy resistance. Schematic representation of cellular interactions within the hypoxic TME. Cancer cells are surrounded by various cells, including Treg, CTLs, NK cells, TAM, TAN, MDSCs, etc. CTLs and NK cells exhibit PD-1 receptors that interact with PD-L1 expressed by TAM2, MDSCs, and DCs in the hypoxic TME. TAMs can undergo polarization and differentiation influenced via the hypoxic TME. TAM1 exhibits antitumor, while TAM2 promotes tumors. MDSCs release a series of cytokines (b-FGF, IGF-1, IL-10, IL-4, IL-1β, SDF-1, and MCP-1) affecting cancer cell behavior. TGF-β and IL-10 act as regulatory molecules inhibiting CTLs and NK cells, respectively. While the MHC I molecule and tumor antigen facilitate the interaction between cancer cells and CTLs, TAN1, and TAN2, differentiated from TAN, play the roles of inhibiting and promoting cancer cells, respectively. This figure illustrates the complex network of cellular interactions within the hypoxic TME.

**Figure 2 cancers-15-05857-f002:**
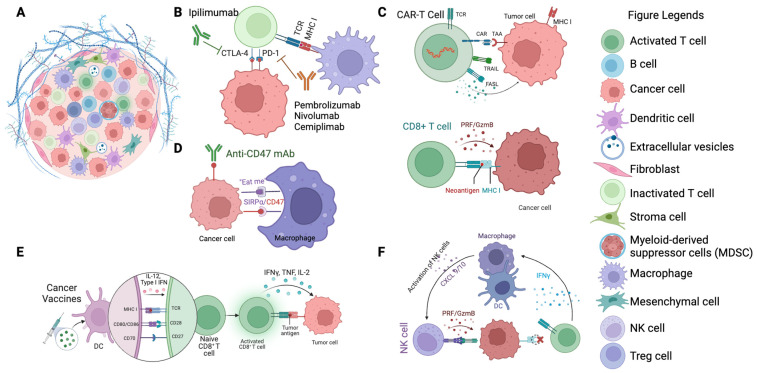
Targets of immunotherapeutic agents in cancer therapy. (**A**) Illustration of the TME featuring cancer cells surrounded by various immune cells and extracellular matrix components. (**B**) Depiction of immune checkpoint inhibitors (ICIs) such as CTLA-4 and PD-1 (e.g., ipilimumab, pembrolizumab, nivolumab, cemiplimab) binding to their respective receptors on T cells, preventing immune evasion by cancer cells. (**C**) Representation of CAR T-cells targeting tumor-associated antigens (TAAs) on cancer cells, triggering cytotoxic responses. (**D**) Macrophage checkpoint inhibition: anti-CD47 mAb blocks the “don’t eat me” signal on cancer cells, promoting their phagocytosis by macrophages. (**E**) Depiction of dendritic cells (DCs) presenting tumor antigens to naïve T cells, leading to their activation and the initiation of an adaptive immune response against cancer cells. (**F**) Illustration of activated NK cells targeting cancer cells, mediated by cytokine signaling (e.g., IFNγ production), which enhances the innate immune response against tumors.

**Table 1 cancers-15-05857-t001:** Overview of pioneering strategies in cancer immunotherapy.

Strategies	Description	Key Components and Benefits	Representative Drugs/Cells/Vaccines	References
Combination Therapies	Integration of several therapeutic modalities to optimize oncological outcomes.	Synergistic modalities enhance response. Versatility against varying tumor behaviors. Potential for prolonged patient benefits.	Anti-NKG2A: Monalizumab, Anti-PD-1: Nivolumab, Pembrolizumab Anti-PD-L1: Atezolizumab, Avelumab, Anti-CTLA-4: Ipilimumab, Durvalumab	[94,95,96,110]
TME	Considers the composite of stromal and immune cells intertwined with signaling pathways. Affects tumor progression and anti-tumor immunity.	Stroma including ECM and fibroblasts; mesenchymal stromal cells; and immune cells such as TAMs, TANs, and Tregs, signaling pathways that influence tumor progression.	Anti-LOXL2: Simtuzumab, anti-hyaluronic acid: PEGPH20, anti-CTGF: Pamrevlumab, anti-Integrin: Cilengitide, ATN-161, MEDI-522, anti-TGF-β: Fresolimumab, etc.	[97,98,127]
Immune Checkpoints (ICIs)	Novel checkpoints open up promising therapeutic possibilities. They modulate immune functions.	Potential checkpoints like TIGIT, TIM-3, and LAG-3 receptors, expanding therapeutic avenues.	Anti-LAG-3 mAbs: Relatlimab, Favezelimab, REGN3767, GSK2831781, LAG525, TSR-033, Relatlimab + Nivolumab, etc. Anti-TIM3: Sabatolimab, spartalizumab.	[127,128]
Adoptive Cell Therapy (ACT)	Capitalizes on an individual’s immune cells. Offers a tailored therapeutic approach.	Precision with techniques like TIL extraction; potential of CAR-T cells provide a tailored therapeutic approach. Enhanced therapeutic results when combined with other modalities.	Tumor-infiltrating lymphocytes (TILs), T cell-receptor-engineered T (TCR-T) cells, natural killer T (NKT) cells	[107,108,109]
Cancer Vaccines	Utilization of neoantigens to boost immune responses targeting tumors.	Innovation with DC vaccines and viral vector vaccines; enhances immune response.	Peptide vaccines: Gardasil^®^, gp96, OSE2101, DSP-7888, etc.; DNA vaccines: HER2, VGX-3100, WT1, P, MA, hTERT, etc. mRNA vaccines: BNT112, BNT113, MAGE-A3, KRAS, etc.; virus-based vaccines: PROSTVAC-V/F, TG4010, BT-001; cell-based vaccines: DC vaccines; GVAX, etc.	[111,112,113,114]

## Data Availability

No new data were created or analyzed in this study. Data sharing does not apply to this article.

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
