# Peer review of "Navigating the Immune Maze: Pioneering Strategies for Unshackling Cancer Immunotherapy Resistance"

_cancers, 2023, doi:10.3390/cancers15245857_

Round 1

Reviewer 1 Report

Comments and Suggestions for Authors

This paper starts as a nice introduction to the increasing problems facing cancer immunotherapeutic interventions in terms of resistance.  However, while it explains at a very high level, some mechanisms by which tumours may be expected to become resistant to therapy (antigens, TME alterations), it does not delve into potential mechanisms for this in any detail.  My major concern is that section 5, recent insights and developments, should be the crux of the paper.  I'd expect this section to be expanded on significantly (perhaps at the expense of the previous section, in which we have an explanation and a table that essentially relay exactly the same information, this could be reduced to the table only).  In section 5 I feel that to improve our understanding of the field, the authors should identify the current understanding of known mechanisms by which immunotherapy contributes to genetic, epigenetic and microbiome changes, with solid examples from the literature.  Without this, this paper does little to add to our existing knowledge and just tells us there is a problem, without delving into currently known causes and examples.

Section 6 provides more nice detail on how we can ensure that these observations are factored into clinical design in order to garner more information, but again more detail could be presented by the authors regarding their use - how can these experimental models influence the treatment design?

Author Response

Subject: Response to Reviewer 1’Comments on Manuscript ID cancers-2728153

Dear Dr. Bousbaa, Dr. Hu, and Dr. Reviewer 1,

We are grateful for the insightful feedback provided by the reviewers and the editorial team, which has significantly contributed to enhancing the quality of our manuscript entitled "Navigating the Immune Maze: Pioneering Strategies for Unshackling Cancer Immunotherapy Resistance." We have thoroughly reviewed and reflected on each comment, incorporating substantial revisions where necessary.  The detailed responses to the reviewers' observations and queries are outlined below, and we trust that these amendments will meet the esteemed reviewers' approval.

Comments from Reviewer 1

This paper starts as a nice introduction to the increasing problems facing cancer immunotherapeutic interventions in terms of resistance.     However, while it explains at a very high level, some mechanisms by which tumors may be expected to become resistant to therapy (antigens, TME alterations), it does not delve into potential mechanisms for this in any detail.

My major concern is that section 5, recent insights and developments, should be the crux of the paper. I'd expect this section to be expanded on significantly (perhaps at the expense of the previous section, in which we have an explanation and a table that essentially relays exactly the same information, this could be reduced to the table only).

In section 5 I feel that to improve our understanding of the field, the authors should identify the current understanding of known mechanisms by which immunotherapy contributes to genetic, epigenetic, and microbiome changes, with solid examples from the literature. Without this, this paper does little to add to our existing knowledge and just tells us there is a problem, without delving into currently known causes and examples.

Section 6 provides more nice detail on how we can ensure that these observations are factored into clinical design in order to garner more information, but again more detail could be presented by the authors regarding their use - how can these experimental models influence the treatment design? 

  1. Lack of detail in mechanisms of resistance:

We acknowledge the reviewer's concern regarding the level of detail in the mechanisms of resistance. To address this, we have expanded our discussion in Section 5 to delve deeper into the genetic, epigenetic, and microbiome changes associated with immunotherapy resistance.  We have included new subsections with specific examples from recent studies that illustrate these mechanisms, providing a richer and more comprehensive analysis.

  1. Expansion of Section 5:

We appreciate Reviewer 1's valuable comments. In line with these constructive suggestions, we have significantly expanded Section 5, including the following subsections:

5.1 Genetic Alterations and Immunotherapy Resistance

5.2 Epigenetic Dynamics and Their Role in Resistance

5.3 The Microbiome's Influence on Immunotherapy Efficacy

5.4 Enhancing Immunotherapy with Oncolytic Viruses

You can find these revisions in the revised version from page 8 to page 10. In addition, we have condensed Section 4 to avoid redundancy, removing textual repetition and retaining the table for succinctness. This allowed us to dedicate additional space to Section 5, where we now present a detailed examination of the current understanding of immunotherapy-induced changes.

  1. Inclusion of Solid Literature Examples:

To substantiate our discussion on how immunotherapy contributes to resistance via genetic, epigenetic, and microbiome alterations, we have added several concrete examples from the literature in Section 5. These examples have been carefully selected to highlight the most recent and relevant findings in the field, providing the reader with a clear understanding of the current state of research.  You can find these examples in the revised version from page 8 to page 10.

  1. Detail on Experimental Models in Section 6:

We appreciate Reviewer 1's request for more detail on the influence of experimental models on treatment design. Section 6 now includes a more explicit discussion of how patient-derived organoids (PDOs) and patient-derived xenograft (PDX) models are used in preclinical studies to tailor and refine treatment strategies.  We have also provided case studies where these models have informed clinical decision-making and led to improved patient outcomes.

We believe that the revisions made to our manuscript have addressed the reviewer's comments comprehensively and have substantially improved the quality and impact of our work. We are confident that these changes will satisfy the concerns raised and make a significant contribution to the literature.

Warm regards,

Wenxue Ma

Wenxue Ma, MD, PhD  

Research Professor, Physician Scientist, Director of Drug Discovery

University of California San Diego

La Jolla, CA, 92093-0695, USA

E-mail: wma@health.ucsd.edu

AE, Frontiers in Immunology, Frontiers in Oncology

Reviewer 2 Report

Comments and Suggestions for Authors 1. Authors should explain the possible ways to overcome the existing challenges with the immunotherapies. 2. Authors should mention the drugs which can causes the immunotherapy resistance 3. Authors should mention the name of the immunotherapeutic agents where ever immunotherapy or related statement has been stated. For example, Page 2, Line 88. 4. Draw a figure for the immunotherapeutic agents and its targets. 5. Mention the drug examples in the table in a separate column Comments on the Quality of English Language

Nil

Author Response

Subject: Response to Reviewer 2’s Comments on Manuscript ID cancers-2728153

Dear Dr. Bousbaa, Dr. Hu, and Dr. Reviewer 2,

We are grateful for the insightful feedback provided by the reviewers and the editorial team, which has significantly contributed to enhancing the quality of our manuscript entitled "Navigating the Immune Maze: Pioneering Strategies for Unshackling Cancer Immunotherapy Resistance." We have thoroughly reviewed and reflected on each comment, incorporating substantial revisions where necessary.  The detailed responses to the reviewers' observations and queries are outlined below, and we trust that these amendments will meet the esteemed reviewers' approval.

Comments from Reviewer 2

  1. Authors should explain the possible ways to overcome the existing challenges with immunotherapies.

We appreciate Reviewer 2's constructive feedback and the opportunity to elaborate on the potential solutions to the challenges facing immunotherapy. To overcome these challenges, a multifaceted approach is necessary. This includes the development of combination therapies that synergize different mechanisms of action to tackle the complex tumor microenvironment and resistance pathways.  We are also exploring the personalization of treatment through precision medicine, utilizing genomic and proteomic profiling to tailor therapies to individual patient tumor profiles.

Furthermore, we are investigating the modulation of the immune system itself, such as enhancing T-cell activity with checkpoint inhibitors and adopting novel agents that can activate immune responses more effectively.  The role of the microbiome in cancer therapy is also under examination, with the potential to manipulate microbial composition to improve immunotherapy outcomes.

Another promising avenue is the use of oncolytic viruses, which can selectively target and kill tumor cells while stimulating an immune response.  There is also a growing interest in adoptive cell transfer therapies, including TILs and CAR-T cell therapies, which provide a more direct attack on cancer cells.

Additionally, overcoming resistance can be facilitated by the identification and monitoring of predictive biomarkers, which can signal the need for adjustments in therapy before clinical resistance becomes apparent.  Finally, research into resistance mechanisms is paving the way for the development of new drugs that can prevent or overcome resistance, ensuring that immunotherapy remains a powerful tool in the fight against cancer."

All the above key points have been integrated into the new section 4 and new section 5 in the revised version. 

  1. Authors should mention the drugs which can cause the immunotherapy resistance. 

Thank you very much for your suggestion. We have included a new subsection titled 'Medications Contributing to Resistance' as section 4.6 from page 5 to page 6, which details specific drugs known to potentially induce resistance to immunotherapy.

  1. Authors should mention the name of the immunotherapeutic agents wherever immunotherapy or related statement has been stated. For example, Page 2, Line 88.

In lines 95-96 on page 2 of the revised text, we have elaborated on immunotherapy, mentioning specific agents such as nivolumab (a PD-1 inhibitor) and ipilimumab (a CTLA-4 inhibitor). Immunotherapy represents a groundbreaking shift in cancer treatment, showing promising results across various malignancies, including melanoma, non-small cell lung cancer, and renal cell carcinoma.

  1. Draw a figure for the immunotherapeutic agents and their targets

Thank you for your suggestion. We have included a figure titled "Targets of Immunotherapeutic Agents in Cancer Therapy" on page 6 in the revised version.

  1. Mention the drug examples in the table in a separate column.

Thank you for your suggestion. We have incorporated an additional column into Table 1, which now includes representative drugs, cells, and cancer vaccines, complete with updated and pertinent references. You can locate the updated table on page 7.

We believe that the revisions made to our manuscript have addressed the reviewer's comments comprehensively and have substantially improved the quality and impact of our work. We are confident that these changes will satisfy the concerns raised and make a significant contribution to the literature.

Warm regards,

Wenxue Ma

Wenxue Ma, MD, PhD  

Research Professor, Physician Scientist, Director of Drug Discovery

University of California San Diego

La Jolla, CA, 92093-0695, USA

E-mail: wma@health.ucsd.edu

AE, Frontiers in Immunology, Frontiers in Oncology 

Round 2

Reviewer 1 Report

Comments and Suggestions for Authors

The authors have submitted a much improved version of the manuscript that delves into more detail that the previous version, as requested.

Further comments are below;

1. In the intro, there is a lot of repetition in the last two paragraphs, please address/merge.

2. Section 4 is entitled 'pioneering strategies to overcome resistance' but is more accurately more of a brief summary of immunotherapy strategies, not directed at discussing overcoming resistance.  Either rename and expand a little on each section, or describe how these strategies are employed to overcome resistance.  In addition, I would move the OVT section into here from section 5 as this is a more fitting place for it.

3. Section 5 is much better, I would just ask for more specifics on when/which treatments have been shown to induce hypermethylation/histone modifications and the impact it had on further treatment.

4. I still think the PDX model section would benefit from a more complete review, in the comment that PDX studies have lead directly to clinical trials and treatment adjustments, can the authors be more explicit on how.

Comments on the Quality of English Language

The English is good, only minor grammatical errors spotted.

Author Response

Dec. 07, 2023

Subject: Response to Reviewer 1’s Round 2 Comments on Manuscript ID cancers-2728153

Dear Dr. Bousbaa, Dr. Hu, Dr. Reviewer 1, and Ms. Yang

We wish to express our gratitude for the valuable feedback provided by both the reviewers and the editorial team. This feedback has played a pivotal role in elevating the quality of our manuscript titled "Charting a Course Through the Immune Maze: Innovative Approaches to Overcoming Resistance in Cancer Immunotherapy." We have diligently examined and pondered over each comment, making substantial revisions wherever deemed necessary. In the following sections, we present comprehensive responses to the reviewers' remarks and questions, with the sincere hope that these revisions will align with the expectations of our esteemed reviewers.

Round 2_cancers-2728153

Reviewer 1 01 Dec 2023 12:39:39

The authors have submitted a much improved version of the manuscript that delves into more detail that the previous version, as requested.

Further comments are below;

  1. In the intro, there is a lot of repetition in the last two paragraphs, please address/merge.

Response: Thank you for bringing this to our attention. We have addressed the concern by consolidating the content and eliminating redundancy in the introduction. The revised text is now more concise and can be reviewed in the tracked changes between lines 59 and 75.

  1. Section 4 is entitled 'pioneering strategies to overcome resistance' but is more accurately more of a brief summary of immunotherapy strategies, not directed at discussing overcoming resistance.  Either rename and expand a little on each section, or describe how these strategies are employed to overcome resistance.  In addition, I would move the OVT section into here from section 5 as this is a more fitting place for it.

Response: We are grateful for the insightful feedback provided.  After reviewing the content, we concur with your assessment regarding the placement of the Oncolytic Viruses Therapy (OVT) section.  Accordingly, we have relocated the original Section 5.4 to Section 4, recognizing that it aligns more appropriately with the focus on overcoming resistance.

Additionally, we acknowledged the overlap between the contents of the former Sections 4.6 and 4.8.  To enhance clarity and cohesiveness, these sections have been consolidated into a single subsection in the updated manuscript (Version 2). This newly merged section is now presented as Section 4.7, entitled "Navigating Medication-Induced Resistance in Immunotherapy," which we believe succinctly encapsulates the essence of the combined content.

These modifications have been reflected in the revised manuscript, and we invite you to examine the changes, particularly from lines 221 to 281, where the relevant adjustments have been tracked for your convenience.

  1. Section 5 is much better, I would just ask for more specifics on when/which treatments have been shown to induce hypermethylation/histone modifications and the impact it had on further treatment.

Response: Thank you for your constructive comments. In response, we have revised the discussion on the specific treatments known to induce hypermethylation and histone modifications, as well as their implications for subsequent immunotherapy. We have incorporated these updates into the revised manuscript to ensure a more comprehensive understanding of how epigenetic therapies intersect with immunotherapy and the subsequent clinical implications. These revisions are reflected in the new sections 5.2 and 5.3 of the manuscript.

  1. I still think the PDX model section would benefit from a more complete review, in the comment that PDX studies have lead directly to clinical trials and treatment adjustments, can the authors be more explicit on how.

In response to reviewer 1's comments regarding the need for a more detailed review of how PDX models have led to clinical trials and treatment adjustments, we have expanded on the specific mechanisms by which these models have informed clinical decisions. See the revised section 6 which is highlighted with tracking in the new version.

We believe that the revisions made to our manuscript have addressed the reviewer's comments comprehensively and have substantially improved the quality and impact of our work. We are confident that these changes will satisfy the concerns raised and make a significant contribution to the literature.

Warm regards,

Wenxue Ma

Wenxue Ma, MD, PhD  

Research Professor, Physician Scientist, Director of Drug Discovery

University of California San Diego

La Jolla, CA, 92093-0695, USA

E-mail: wma@health.ucsd.edu

AE, Frontiers in Immunology, Frontiers in Oncology 

Round 3

Reviewer 1 Report

Comments and Suggestions for Authors

This version seems much improved and offers a more complete review of the field.  Minor comment in lines 372-373, this sentence needs to be addressed to make sense.